# What Is Different in Acute Hematologic Malignancy-Associated ARDS? An Overview of the Literature

**DOI:** 10.3390/medicina58091215

**Published:** 2022-09-03

**Authors:** Mihail Cotorogea-Simion, Bogdan Pavel, Sebastian Isac, Teodora Telecan, Irina-Mihaela Matache, Anca Bobirca, Florin-Teodor Bobirca, Razvan Rababoc, Gabriela Droc

**Affiliations:** 1Department of Anesthesiology and Intensive Care I, Fundeni Clinical Institute, 022328 Bucharest, Romania; 2Department of Physiology, Faculty of Medicine, Carol Davila University of Medicine and Pharmacy, 020021 Bucharest, Romania; 3Department of Urology, Iuliu Hatieganu University of Medicine and Pharmacy, 400012 Cluj-Napoca, Romania; 4Department of Urology, Municipal Hospital, 400139 Cluj-Napoca, Romania; 5Department of Rheumatology, Dr. I. Cantacuzino Hospital, 073206 Bucharest, Romania; 6Department of General Surgery, Dr. I. Cantacuzino Hospital, 073206 Bucharest, Romania; 7Department of Internal Medicine II, Fundeni Clinical Institute, 022328 Bucharest, Romania

**Keywords:** acute hematologic malignancy, acute respiratory distress syndrome, pneumonia, leukostasis, diffuse alveolar hemorrhage, engraftment syndrome, radiation recall pneumonitis, transfusion-related acute lung injury

## Abstract

*Background and Objectives*: Acute hematologic malignancies are a group of heterogeneous blood diseases with a high mortality rate, mostly due to acute respiratory failure (ARF). Acute respiratory distress syndrome (ARDS) is one form of ARF which represents a challenging clinical condition. The paper aims to review current knowledge regarding the variable pathogenic mechanisms, as well as therapeutic options for ARDS in acute hematologic malignancy patients. *Data collection*: We provide an overview of ARDS in patients with acute hematologic malignancy, from an etiologic perspective. We searched databases such as PubMed or Google Scholar, including articles published until June 2022, using the following keywords: ARDS in hematologic malignancy, pneumonia in hematologic malignancy, drug-induced ARDS, leukostasis, pulmonary leukemic infiltration, pulmonary lysis syndrome, engraftment syndrome, diffuse alveolar hemorrhage, TRALI in hematologic malignancy, hematopoietic stem cell transplant ARDS, radiation pneumonitis. We included relevant research articles, case reports, and reviews published in the last 18 years. *Results:* The main causes of ARDS in acute hematologic malignancy are: pneumonia-associated ARDS, leukostasis, leukemic infiltration of the lung, pulmonary lysis syndrome, drug-induced ARDS, radiotherapy-induced ARDS, diffuse alveolar hemorrhage, peri-engraftment respiratory distress syndrome, hematopoietic stem cell transplantation-related ARDS, transfusion-related acute lung injury. *Conclusions:* The short-term prognosis of ARDS in acute hematologic malignancy relies on prompt diagnosis and treatment. Due to its etiological heterogeneity, precision-based strategies should be used to improve overall survival. Future studies should focus on identifying the relevance of such etiologic-based diagnostic strategies in ARDS secondary to acute hematologic malignancy.

## 1. Introduction

Hematologic malignancies are a diverse group of pathologies, which affect an estimated 548.8 per 100,000 people, comprising roughly 20% of all types of neoplasia [1,2]. They can be divided, according to the most common subtypes, into leukemias, Hodgkin’s lymphomas, non-Hodgkin’s lymphomas, multiple myeloma, myelodysplastic syndromes, and myeloproliferative disorders, each with its origin, pathogenic mechanisms, incidence, burden on health, and mortality. In 2018, leukemia had an incidence of 407,000 cases and claimed 309,000 lives [3]. There is a marked heterogeneity even within this subpopulation. For example, the age-standardized incidence rate (ASIR) of leukemia as a whole decreased by approximately 9% between 1990 and 2017, mostly due to reductions in acute lymphocytic leukemia and chronic myeloid leukemia, whereas the values for chronic lymphocytic leukemia and acute myeloid leukemia increased [3].

The therapeutic options for hematologic malignancies can be divided into chemotherapy, radiotherapy, immunotherapy, targeted therapy, and hematopoietic stem cell transplantation (HSCT) [4,5,6]. Immunotherapy uses compounds that augment the natural defense capabilities of the organism, such as increasing interferon production or enhancing antibody-dependent cellular cytotoxicity [5]. Targeted therapy refers to drugs which disrupt the metabolic pathways specific to the cancer cell and its genetic abnormalities, discriminately targeting rapidly dividing cells [4].

The most common cause for ICU admission among patients with hematologic malignancies is acute respiratory failure (ARF), which also accounts for most non-relapse deaths in such cases [2]. Furthermore, acute hematologic malignancies are particularly prone to developing respiratory complications [1]. According to the literature, the main causes of ARF could be related to the injury of the lung parenchyma, pneumonia with opportunistic agents, chemo- or radiotherapy-induced ARF, transfusion-related acute lung injury, leukemic infiltration of the lung, tumor lysis pneumopathy, pulmonary alveolar proteinosis, and diffuse alveolar hemorrhage or engraftment syndrome. Other causes include chest wall and pleura damage, thromboembolic events, or neuromuscular origin (paraneoplastic syndromes, metabolic encephalopathy, sedative-induced) [2].

Acute respiratory distress syndrome (ARDS) is a particularly relevant entity, due to its high mortality. ARDS is defined, according to the Berlin criteria, by the presence of: (i) hypoxemia (PaO_2_/FiO_2_ ≤ 300 mmHg with PEEP or CPAP ≥ 5 cm H_2_O), (ii) respiratory failure not completely attributable to cardiac failure or fluid overload, (iii) onset within one week of exposure to a known risk factor, and (iv) bilateral opacities on chest X-ray or CT scan which cannot be fully explained by atelectasis, nodules, or pleural effusions [7]. In patients with hematologic malignancy, ARDS can result from either direct or indirect lung injury. Both mechanisms follow, however, a similar pattern. First, damage to pulmonary epithelial and endothelial cells leads to inflammation, apoptosis, and necrosis of alveolar type I and II cells, and to increased permeability of the alveolar-capillary membrane, causing fluid exudation and hemorrhage in the alveoli [7,8]. Then, during the proliferative phase, the barrier’s integrity is reestablished, and the excess fluid is reabsorbed [7,8]. Lastly, as an optional occurrence, fibrotic tissue begins to appear in the lung, with unfavorable consequences in terms of mortality or recovery (Figure 1) [7,8].

ARDS management is largely supportive [7]. Most of the ARDS therapies target optimizing mechanical ventilation and obtaining an appropriate end-expiratory pressure, to prevent alveolar atelectrauma and additional fluid buildup in the alveoli, while simultaneously keeping plateau pressure at acceptable levels, to avoid barotrauma [7]. Maintaining a minimal change in flow variability between respiratory cycles could prevent ergotrauma [9,10]. Regarding the positive end-expiratory pressure (PEEP), meta-analyses have come up with conflicting results in terms of the effectiveness of higher PEEP compared to lower values, with discernible benefits only in selected patient subgroups [11,12,13]. One therapeutic intervention widely accepted is ventilation using low tidal volumes (4–6 mL/kg predicted body weight), which appears to be effective regardless of the putative ARDS triggering factor, accompanied by reduced circulating interleukin-6 levels [7,11,12,13]. The use of prone positioning aims to improve matching between ventilation and perfusion by taking advantage of gravity and intrathoracic organ repositioning, with promising results in severe ARDS in terms of oxygenation, and reduced mortality [7,14,15]. Neuromuscular blocking agents administered for short periods also improve oxygenation, while reducing mortality, barotrauma, and overall requirement for mechanical ventilation in severe ARDS cases [16]. These drugs also reduce the level of circulating inflammatory mediators [7,17]. Other specialized interventions have presented some promising results and are subject to further research, such as inhaled prostaglandins or nitric oxide, CO_2_ removal strategies such as CO_2_ removal filters attached to dialysis machines, and venovenous extracorporeal membrane oxygenation [18,19,20,21].

Studies attempting to prove the potential usefulness of targeted therapeutic measures in reducing mortality have largely come up with negative results [17,22,23,24]. With the dawn of personalized medicine, this has been attributed to the heterogeneity of ARDS, in terms of its clinical, physiologic, biochemical, and radiologic features, which also leads to variable responses to different therapeutic options, especially in patients with hematologic malignancy [25,26,27].

When referring to clinical classification, differences were noted when determining the cause of thoracic stiffness, with direct ARDS causing increased elastance in the lung parenchyma, whereas indirect ARDS exhibits higher elastance in the chest wall [26]. Within the broad group of indirect risk factors, higher mortality rates were noted in sepsis- vs. non-sepsis ARDS, further supporting the idea of a need to tailor the management to each individual case, instead of applying a cookie-cutter treatment to all patients [25]. A higher mortality rate was also noticed in cases of late onset ARDS (days 3–4 after the triggering injury), while early-onset patients exhibited higher plasma levels of biomarkers, suggestive alveolar–endothelial barrier disruption [26].

Biochemical inhomogeneity in ARDS patients stems from the blood and bronchoalveolar lavage fluid (BALF) profiles of various inflammatory endothelial and epithelial markers [26,28]. The main biochemical changes in ARDS are displayed in Figure 2.

The phenotypes most relevant and potentially useful for clinical practice are the so-called “hyperinflammatory” and “hypoinflammatory” phenotypes. The hyperinflammatory phenotype exhibits higher levels of circulating proinflammatory interleukins (IL-6, IL-8, IL-10), PAI-1, soluble receptors for TNF-α and advanced glycation end-products, and lower levels of plasma C protein and surfactant proteins in the BALF (Figure 2) [25,28]. This classification is relevant to clinical practice in terms of estimating patient mortality and therapeutic strategy. Thus, a study by Famous et al. found that hyperinflammatory phenotype cases benefit from a liberal fluid therapy strategy, while hypoinflammatory ARDS patients had a lower mortality when assigned to a fluid-conservative approach [29]. Additionally, treatment plans might be influenced by which phenotype the patient fits in. The HARP-2 trial observed that those with a hyperinflammatory phenotype had a better 28-day survival when receiving 80 mg Simvastatin [30]. The same could not be said about a 40 mg loading/20 mg maintenance dose of Rosuvastatin. This could come down to differences in drug bioavailability and intracellular concentrations [30]. Moreover, tailoring ARDS management according to the patients’ inflammatory phenotype in hematologic malignancy is challenging due to the malignant cellular clones’ abnormal response to inflammation.

From a radiological standpoint, ARDS can be divided into focal or non-focal [25]. Non-focal ARDS appears on thoracic CT scans as diffuse alveolar opacity, and is correlated with higher mortality, while focal ARDS have patches of parenchyma with loss of aeration, particularly in the lower lobes and dependent regions [25,26]. This dichotomy influences therapeutic management and expected mortality. In cases of focal ARDS, resorting to a low PEEP and normal tidal volume strategy, combined with prone positioning to optimize ventilation/perfusion ratio, might have beneficial effects [25,26]. This observation relies on the idea that using high PEEP in lungs with inhomogeneous compliance would lead to overdistension of already compliant areas, while the damaged zones would not benefit from the greater pressure [25,26]. On the other hand, non-focal ARDS benefits from recruitment maneuvers, meant to expand the collapsed alveoli, combined with higher PEEP to keep them open [25,26].

This review focuses on the main causes of ARDS in hematologic patients, considering the challenges of a precision-based approach in associated blood malignancy.

## 2. Data Collection

We used, for the narrative review, articles appearing in various databases such as PubMed or Google Scholar until June 2022. We used as key words ARDS in hematologic malignancy, pneumonia in hematologic malignancy, drug induced ARDS, leukostasis, pulmonary leukemic infiltration, pulmonary lysis syndrome, engraftment syndrome, diffuse alveolar hemorrhage, TRALI in hematologic malignancy, hematopoietic stem cell transplant ARDS, radiation pneumonitis, providing a total number of 11,261 articles. We included research articles, case reports and reviews referring to various respiratory complications in hematologic malignancy. We added relevant articles published in the last 19 years.

## 3. ARDS in Acute Hematologic Malignancy-Specific Causes

### 3.1. Pneumonia-Associated ARDS

Owing to the disease itself, as well as its therapeutic options, patients with acute hematologic malignancy find themselves immunosuppressed, particularly neutropenic (<500 neutrophiles/mm^3^) [31]. Febrile neutropenia patients suffer most often from pulmonary complications, with a rate of 15–20% [31]. Pneumonia occurs in 13–31% of patients undergoing induction chemotherapy and up to 80% of those receiving hematopoietic stem cell transplants [31]. Other studies revealed that ARDS secondary to pneumonia is the main reason for ICU admission in cancer patients [32,33]. The study by Azoulay et al. also noted that, of all cases of pneumonia-induced ARDS, 58.16% were of bacterial origin and 32.18% were fungal, with 9.67% of infections caused by *Pneumocystis jirovecii* [32]. A review by Evans and Ost found that mortality among leukemia patients ranges between 25 and 80%, standing at 90% in the case of stem cell transplantation patients [33]. While most patients with hematologic malignancy tend to have frequent contact with hospitals, one must remember the possibility of community-acquired pneumonia in neutropenic individuals. Commonly incriminated bacterial agents are *Streptococcus pneumoniae* (especially in patients with dysfunctional humoral immunity) and *pyogenes*, *Staphylococcus aureus*, *Pseudomonas* spp., non-fermentative Gram-negative bacilli (including *Moraxella* and *Stenotrophomonas*), *meningococcus*, and atypical germs, such as *Mycoplasma, Chlamydophila*, and *Legionella* [2,33]. Influenza, parainfluenza, and adenoviruses represent the bulk of viral community-acquired pneumonia [2,33]. Nosocomial pneumonia (healthcare-associated, hospital-acquired, and ventilator-associated, with similar causative microbes) is most often produced by multidrug resistant bacteria, along with *Enterobacteriaceae*, *Nocardia* spp., and Mycobacteria—both *M. tuberculosis* and atypical [33]. Other etiological agents are fungi (mostly *Aspergillus* spp., with mucormycosis on the rise in recent years), *Pneumocystis*, and viruses (respiratory syncytial virus, metapneumovirus, varicella zoster and human herpes virus, cytomegalovirus, SARS-CoV-2) [32,33]. A secondary analysis of the Global Initiative for MRSA Pneumonia (GLIMP) database discovered that hematological malignancies were significantly more often associated with community-acquired pneumonia caused by fungi and non-influenza viruses [34].

Cytomegalovirus is a member of the herpesvirus family, and its prevalence in the general population exceeds 80% in Europe and North America, being close to omnipresent in Africa and Asia, owing to its multiple transmission paths (blood, saliva, breast milk, sexual contact) and the persistent infection it causes [35]. Cytomegalovirus uses macrophages and CD34+ cells, which include hematopoietic stem cells as reservoirs [35,36]. CD34+ cells include those cells used in stem cell transplant procedures [36]. Infection reactivation and progression to clinical disease depends on the proper reconstitution of the various T-cell subtypes following HSCT [36]. Unfortunately, the fact that the process is dependent on proper thymus function, which is impaired in hematologic malignancies, means that physiological proportions of T-cell subtypes (mainly the CD4:CD8 ratio) cannot be reached, leading to compromised anti-cytomegalovirus protection [36].

Aspergillus can disseminate in the lungs either through the blood vessels, causing infarction in the surrounding tissues, or by way of the airways, being frequently incriminated in patients with hematological malignancies [37]. However, the prevalence of infection does not strictly correlate with the depth of the immune dysfunction, which suggests that a genetic component might also be involved [38]. Multiple studies have investigated the effects which the single-nucleotide polymorphisms (SNP) in genes coding for components of the immune system have on the risk of developing invasive pulmonary aspergillosis. While many of the investigated polymorphisms had no statistically significant impact on invasive aspergillosis rates in the investigated stem cell transplant recipients, there were some which had deleterious consequences on the host’s defensive capabilities, such as: (1) Toll-like receptor (TLR) 1, 3, 4, 5, and 6; (2) IL-4 receptor; (3) Plasminogen; (4) Vascular endothelial growth factor (VEGF); and (5) IL-8 [38,39,40,41,42,43,44]. TLR-4 and 5 were unexpected results, since they bind lipopolysaccharides, usually found in bacterial walls, and of which Aspergillus has none [38]. The IL-4 receptor polymorphism (rs2107356) has also been associated with multiple myeloma, gastric cancer, thymoma-associated myasthenia gravis, and graft dysfunction after kidney transplantation [45,46]. Plasminogen bound to Aspergillus active conidia, potentially facilitating its entry in the organism and enhancing tissue damage [42]. The relationship between angiogenesis and aspergillosis appeared to be bidirectional—VEGF inhibition increases the susceptibility to the disease, with the fungus producing toxins capable of inhibiting angiogenesis [43,44]. However, the products of some SNPs had favorable interactions with Aspergillus, presenting protective effects against the infection. SNP-induced interferon-γ overproduction enhances fungicidal capabilities in macrophages [39]. IL-23 receptor hinders Aspergillus clearance by neutrophils and leads to chronic inflammation due to the IL-23/Th17 pathway [47]. IL-17 promotes fungal germination through its inhibition of indoleamine 2,3-dioxygenase, which also plays a role in subverting the function of T-helper type 1 cells [40,47]. T-helper type 1 cells are responsible for the production of interferon-γ. Thus, decreasing IL-17 production through the mutant IL-23 receptor leads to a lower likelihood of invasion by *Aspergillus*, as well as to protection against graft-versus-host disease (GVHD) in stem cell transplant recipients [40,47,48]. Lower levels of the anti-inflammatory IL-10 lead to resistance to invasive aspergillosis [49]. Finally, IL-12 has also a protective effect, by involving the T-helper type 1 response [49].

Over the past 2 years, SARS-CoV-2 has proven to be a serious challenge to healthcare systems worldwide, having infected over 270 million people and claiming 5.3 million lives worldwide by the end of 2021 [50]. Thus, it became unavoidable that some of the hematologic malignancy patients would contract it as well. A small observational study in China comparing COVID-19 rates among healthcare providers and hospitalized patients with hematological malignancy found no difference. However, ARDS rates were higher, as was mortality, among patients with hematologic neoplasia [51]. This finding is in accordance with the “cytokine storm” theory of COVID-19 pathogenesis, wherein the virus potently activates T-helper type 1 lymphocytes, leading to increased production of IL-6, one of the inflammatory markers incriminated in the onset of ARDS [52]. However, Suárez-Garcia et al. have shown that immunosuppression tends to be paradoxically associated with worse outcomes in COVID-19 pneumonia (ARDS, ICU admission, death) [53]. When compared to other viruses, such as influenza, SARS-CoV-2 appears to be more capable of inducing potentially fatal severe inflammatory responses in hematological malignancy patients [54].

Pneumonia-induced ARDS management includes, beyond standard ARDS supportive measures, antimicrobial therapy with the goal of eradicating the causative agent. The most recent ECIL guidelines recommend an escalation/de-escalation empirical approach. The choice of antimicrobials should be based on the risk of the patient having contracted a resistant germ, as well as the resistance profile of the commonly encountered microbes in the local healthcare setting [55]. For patients without prior resistant pathogen infection or risk of complicated disease evolution, empiric therapy consists of initial monotherapy (piperacillin/tazobactam 4/0.5 g IV q6h or ceftazidime 2 g IV q8h or cefepime 2 g IV q8h or q12h), for a duration of 7–8 days [55,56]. Should this prove ineffective, with deteriorating patient status or proven microbial resistance, the regimen should be changed. The recommended antibiotic regimen must provide broader coverage: carbapenems (meropenem 1 g IV q8h or imipenem 500 mg IV q6h) or antipseudomonal beta-lactams combined with aminoglycosides (e.g., ceftazidime 2 g IV q8h + amikacin 20 mg/kg IV q24h) or with colistin (e.g., ceftazidime 2 g IV q8h + colistin 9 million UI IV loading dose, then 3 million UI IV q8h) [55,56]. Vancomycin should be added if the hospital reports high rates of methicillin-resistant *S. aureus* [55,56]. De-escalation strategies follow the inverse steps and are recommended when bacteriological results are available, in case the patient had been infected or colonized with resistant germs, exhibits signs of unfavorable evolution (hypotension, shock), or if the healthcare center often deals with multidrug-resistant germs [55]. For de-escalation, the course of antibiotic treatment lasts for 14 days [55]. Should the patient suffer from *Legionella* pneumonia, treatment choice is 500 mg levofloxacin IV q12h for 21 days [2,56].

In case of viral pneumonia, etiological treatment is available for influenza viruses, adenoviruses, respiratory syncytial virus, and cytomegalovirus. Influenza virus often responds to oseltamivir 75 mg p.o. q12h or, alternatively, zanamivir [2]. For adenovirus, the treatment of choice is cidofovir in varying reported doses, mostly 5 mg/kg IV weekly for 3 weeks, or 1 mg/kg IV 3 times weekly for 3 weeks in patients with underlying renal dysfunction [57]. For respiratory syncytial virus, available treatment options include ribavirin (aerosols 2 g over 2 h q8h, or 6 g over 18 h, 7–10 days) and RSV immunoglobulins [2]. For cytomegalovirus, ganciclovir 5 mg/kg IV q12h or foscarnet 60 mg/kg IV q8h are recommended [2].

For fungal lung infections, the regimens vary based on the causative agent. Invasive pulmonary aspergillosis requires voriconazole 6 mg/kg i.v. on day 1, then 4 mg/kg q12h (p.o. if renal dysfunction) or amphotericin B (1 mg/kg qd if deoxycholate or 3 mg/kg qd if liposomal, depending on kidney function), in case of no response to voriconazole [2,58]. Mucormycosis exhibits reduced susceptibility to voriconazole [2]. Amphotericin B 5–10 mg/kg qd should be administered until resolution of either clinical or radiologic features, or of the immunosuppression [2]. Trimethoprim-sulfamethoxazole 5 mg/kg i.v. or p.o. (nearly 100% bioavailability) q8h or 3.75 mg/kg q6h for 21 days is the treatment of choice for *Pneumocystis jirovecii* [2]. Alternatively, clindamycin/primaquine or pentamidine can be used for severe cases [2,58]. Additionally, patients infected with *P. jirovecii* are prone to developing pneumothorax and pneumomediastinum, which is challenging for mechanically ventilated patients [59,60].

### 3.2. Leukostasis, Leukemic Infiltration of the Lung, Pulmonary Lysis Syndrome

*Leukostasis* is mostly a complication of myeloid leukemias (acute myelomonocytic or monocytic, and chronic during the blast crisis), especially those with leukocyte counts over 50,000/mm^3^, although the degree of hyperleukocytosis does not necessarily correlate with the severity of the symptoms [61,62]. Leukostasis consists of white blood cell build-up inside small vessels, not only in the lungs, but also brain and heart, among other places, which explains the symptoms associated with this condition (acute respiratory failure, acute myocardial infarction, right ventricular overload, headaches, dizziness, tinnitus, coma, intracranial bleeding, peripheral ischemia, mesenteric infarction, priapism etc.) [61,63,64]. Leukostasis occurs not just due to increased viscosity and low flow in the pulmonary circulation, but also because of cytokines (mostly IL-1 and TNF-α) released by the pathologic cells [61,64]. The cytokine buildup leads to increased expression of adhesion molecules on endothelial cells (such as selectins and ICAM-1), leukocyte aggregation and activation, and secretion of matrix metalloproteinases, causing endothelial damage, increased vascular permeability, and subsequent extravasation of fluid, blood, and leukemic cells [61,64,65]. This migration from the intravascular to the interstitial and alveolar spaces is the basis for the radiologic opacities and the hypoxic respiratory failure that constitute hallmarks of ARDS [64]. However, the aforementioned hypoxemia has yet another causative mechanism—the occlusion of pulmonary capillary vessels, mimicking a pulmonary embolism, which explains how patients with histologically diagnosed leukostasis can be hypoxemic, yet show no abnormalities on chest X-rays [64].

Leukemic infiltration of the lung entails blasts building up in the pulmonary extravascular space, without any other discernible causes (infectious, hydrostatic, or otherwise) [66]. Leukostasis causes migration of leukemic cells into the interstitium. Thus, leukostasis and leukemic pulmonary infiltration are not two separate entities, but rather two sides of the same coin [64]. The two seem to occur more often in myeloid leukemia patients, but infiltration, unlike leukostasis, is even less correlated with hyperleukocytosis. However, it should be suspected in patients with a blast ratio exceeding 40% of total peripheral blood leukocytes [61,67]. The symptomatology is rather sparse, with the patient usually complaining about cough, fever, and dyspnea [61]. Imagistic findings include thickening of the interlobular septa or the bronchovascular bundles, as well as non-systematized “ground-glass” opacities [61,64,66].

Management of ARDS in the case of leukostasis and leukemic infiltration includes, beyond general supportive measures, therapies meant to reduce blood viscosity and deplete the number of leukocytes in the pulmonary vasculature and parenchyma [61]. Thus, patients should receive generous infusions of isotonic intravenous solutions, while avoiding blood transfusions for as long as the patient’s clinical status does not call for urgent action [61]. For leukodepletion, clinicians can resort to either chemotherapy, in the shape of hydroxyurea, or leukapheresis, a process in which the patient’s blood is run through a device containing a centrifuge, which mechanically separates cellular elements from plasma, before being fed back into the blood vessels [61,68]. In case of rapid degradation of clinical status, leukapheresis should be the preferred option, as it leads to a quicker drop in leukocyte levels [68]. However, in promyelocytic leukemia, this procedure should not be used, for two reasons: due to the disseminated intravascular coagulation which contraindicates apheresis, and because of the usually lower than normal white blood cell count [68,69]. Leukapheresis does not influence the long-term survival [68,70,71,72,73,74].

Hydroxyurea has been used as the mainstay of leukodepletion therapy for many decades, as it effectively lowers the number of leukocytes, with a low occurrence of acute tumor lysis syndrome, albeit over a longer period of time (24–48 h after initiation of therapy) [61,75]. Some studies have proven its usefulness in preventing short-term mortality [61,68,76]. Hydroxyurea acts by inhibiting ribonucleotide reductase, preventing the synthesis of deoxyribonucleotides, and halting the cell cycle in the S phase [77].

Due to the involvement of cytokines in leukostasis, corticosteroids have proven themselves useful in reducing leukemic pulmonary infiltration, mortality, and relapse incidence, as well as improving overall and disease-free survival [62,78]. Corticoids act by binding to cytoplasmic glucocorticoid receptors, which then relocate to the nucleus, exerting effects predominantly by induction of gene transcription (selective acetylation of histones) and increasing the expression of anti-inflammatory products such as IL-10 and IκB-α (inhibitor of NFκB) [79,80]. Conversely, they also bind and inhibit other proteins that act as histone acetyltransferases and activators, but for proinflammatory genes, thus switching them off [79]. Another mechanism relies on destabilization of mRNA molecules that encode for inflammatory proteins [77].

Pulmonary lysis syndrome, also known as acute lysis pneumopathy, occurs after initiation of cytostatic treatment [61,67]. However, it does not owe its effects to its direct mechanism of action, but rather to the massive destruction of tumor cells, which release their cytotoxic contents, producing diffuse alveolar damage or lung hemorrhage [61,67]. The incriminated components include reactive oxygen species, enzymes, and damage-associated molecular patterns (cellular components such as DNA, histones, heat shock proteins, uric acid) [61,81]. White blood cell count appears inconsequential, as cases have been reported in patients with fewer than 50,000 leukocytes per mm^3^ [61]. The clinical presentation is typical for acute respiratory failure, while imaging typically shows bilateral “ground-glass” opacities [74]. Manifestations usually appear within 48 h of induction of therapy, but exceptions were noted by studies by both Azoulay et al. and Kunitomo et al., at 15 and 14 days, respectively [82,83]. Management of pulmonary lysis syndrome-induced ARDS includes corticosteroids and supportive therapy. There has been some debate regarding the usefulness of chemotherapy cessation [2,61,74].

### 3.3. Drug-Induced ARDS

Drugs administered in the treatment of hematological malignancy could lead to ARDS not only through their intrinsic action towards the lung, but also by way of their interaction with the neoplastic cells. The incidence varies from 0.1 to 15% [84].

The pathophysiology of drug-induced ARDS is complex, with incriminated mechanisms ranging from idiosyncratic reactions to anaphylaxis, capillary leak syndrome, or reactive oxygen species and inflammatory cytokine production [85]. The relevant drugs incriminated in lung damage leading to ARDS are: bleomycin, mitomycin-C, cyclophosphamide, gemcitabine, cytarabine, GM-CSF, and vinca alkaloids [86,87,88,89]. Bleomycin and mitomycin-C increase reactive oxygen species production [86,87]. Gemcitabine increases cytokine release [86,87]. Cytarabine has a direct toxic effect [88]. GM-CSF increases neutrophil adhesion to lung endothelium, due to higher expression of glycoproteins, and superoxide production [88]. Vinca alkaloids cause endothelial dysfunction by disrupting the organization of tubulin [89].

Of particular interest is all-trans retinoic acid (ATRA), which is used in acute promyelocytic leukemia, where a chromosomal translocation leads to a change in the function of the retinoic acid receptor [90]. Consequently, the gene responsible for cell maturation and differentiation no longer responds to physiological ATRA doses [90]. Thus, the myeloid cells remain trapped in their promyelocyte stage [90]. Another drug used in acute promyelocytic leukemia is arsenic trioxide, which increases degradation of the mutant receptor in the lysosome [91]. The administration of either of these drugs could lead to retinoic acid syndrome, a particular type of drug-induced ARDS. This is an entity which appears in 2 to 31% of patients treated with such drugs, mostly when treatment consists of these alone, during the induction phase, usually 10 days after the initiation of the treatment [92]. When ATRA binds to the retinoic acid receptor, immature cells are forced to differentiate. This changes the profile of secreted cytokines (IL-1β, IL-6, IL-8), increasing expression of lymphocyte function-associated antigen 1 (a molecule involved in the migration of leukocytes), intercellular adhesion molecule 1, matrix metalloproteinase 9, and cathepsin G [93,94]. These changes increase the vascular permeability and facilitate lung infiltration [93,94]. Some of the cytokines are also involved in altering hemostasis [92,93]. Thus, retinoic acid syndrome-associated ARDS manifests itself as leukemic infiltration of the lung and alveolar hemorrhage. Management of this pathology consists of intravenous dexamethasone (10 mg i.v. q12h), stopping the administration of the incriminated drug, and the addition of a different cytostatic agent in cases of leukocytosis [93].

### 3.4. Radiotherapy-Induced ARDS

Management options for hematologic malignancies go beyond pharmacological means. Radiation therapy is also useful, especially in lymphomas, where irradiating affected lymph nodes in selected patients leads to excellent 5-year survival and relapse rates [95]. Its use also extends to leukemia, but more as prophylactic, post-chemotherapy, or palliative therapy [95]. However, body tissues are also susceptible to radiation damage, with the lungs being the most sensitive of the thoracic organs and radiation pneumonitis occurring at doses as low as 15–16 Gy [95]. The radiation-induced death of endo- and epithelial cells leads to a vicious circle of inflammation, increased vascular permeability, and cytokine release, while infiltrating macrophages amplify tissue damage by producing reactive oxygen and nitrogen species and cytokines [96]. The cytokine milieu varies with the time elapsed since the pulmonary injury [96]. The first 2 weeks are characterized by high levels of TNF-α, IL-1 and -6, fibroblastic and platelet-derived growth factors. On a tissular level, this stage is characterized by vascular congestion and intra-alveolar edema, leukocyte infiltration, and pneumocyte apoptosis [96]. In later stages (about 6–8 weeks after the original insult), TGF-β1 expression increases, while vascular and alveolar linings begin to detach, leading to capillary lumen reduction and thrombi formation, and to alveolar collapse with associated fibrin exudation and hyaline membranes, respectively [96].

Radiation pneumonitis could occur months, even years, after radiotherapy [97,98]. In such patients, the triggering factor was proven to be a round of chemotherapy, although cases have been reported where immunotherapy was incriminated instead [96,99]. The mechanisms involved in radiation recall pneumonitis are still being investigated. However, postulated theories include: (1) constant subliminal inflammatory cytokine secretion; (2) changes in local stem cell function, either increased turnover (which increases their susceptibility to antineoplastic agents) or reduced proliferation; (3) accumulation of the anticancer drug due to local changes in angiogenesis and vascular permeability [98,99]. The severity of symptoms does not appear to be correlated with the time elapsed between radio- and chemotherapy [96]. One could mistake radiation recall for chemotherapy-induced lung damage; however, in the case of radiation recall pneumonitis, the ground-glass opacities and infiltrates conform to the shape of the previously irradiated areas [98].

Treatment of ARDS induced by radiation therapy consists mainly of intravenous corticosteroids [96]. Furthermore, some prophylactic options exist, which dampen the effects radiation has on the lung tissue: pentoxifylline, with its TNF-α and IL-1 suppressing action leading to an improvement in symptoms, and amifostine, which acts by scavenging free radicals and by inducing tissular hypoxia, with protective effects [96,100].

### 3.5. Hematopoietic Stem Cell Transplantation-Related ARDS

While chemo- and radiotherapy have remission and symptom control as their goals, stem cell transplantation has been used with curative effects [101]. There are multiple types of transplantation, but the two most widely used are autologous and allogeneic.

Autologous transplant involves harvesting stem cells from the patient, either directly from the bone marrow or from the blood after marrow stimulation [102]. Then the patient undergoes myeloablative therapy, which destroys the malignant cells, along with their own hematopoietic cells, and has the harvested stem cells reimplanted, in hope that they would resume their function [102]. While considered a curative therapy option, relapse rates remain high, mostly due to the stem cell harvest contamination by neoplastic cells [103].

Allogenic HSCT requires a donor, related to the patient or not, with HLA antigen matching [104]. While the complications and non-relapse mortality rate of allogenic HSCT is worse than that of the autologous one, lower relapse rates offset the difference, leading to similar long-term survival [104]. The benefit of allogenic grafts is an immune reaction mediated by minor histocompatibility antigens, which prevents the subsequent growth of leukemic cells [105]. The minor histocompatibility antigen is usually expressed on cells belonging to the immune system, including the malignant ones [105]. However, the donor cells sometimes react with epithelial cells, which also express such antigens, leading to graft vs. host disease [105]. GVHD occurs due to pre-existent damage to the host tissues, through the underlying disease or the preconditioning chemotherapy, which leads to an elevated state of inflammation in the body, culminating in ARDS [106]. Of note is the occurrence of GVHD in autologous stem cell transplant recipients, in spite of the complete cellular antigen matching [107,108]. The putative mechanism is the loss of self-reactive cell suppression, either through direct regulatory T cell expression inhibition (caused by specific agents, such as thalidomide derivatives), or through poor thymus function owing to cytotoxic therapy [107].

Other mechanisms related to hematopoietic SCT, which led to ARDS, are diffuse alveolar hemorrhage, peri-engraftment respiratory distress syndrome (PERDS), and cryptogenic organizing pneumonia [37].

Diffuse alveolar hemorrhage is an exclusion diagnosis, being defined as lung hemorrhage-induced ARDS in the absence of any infection within 1 week after hematopoietic stem cell transplantation [107]. The diffuse alveolar hemorrhage can last between 1 week and 1 month, during the engraftment period, when neutrophil production increases, causing them to flow towards the pulmonary vasculature [37]. To establish the diagnosis, bronchoalveolar lavages must be performed [109]. The bronchoalveolar lavage must appear increasingly bloody as time passes or contain macrophages which are loaded with hemosiderin in proportion higher than 20% [109]. The initial pulmonary lesion is caused by high-dose radiotherapy, releasing host antigens into the circulation, which are then recognized by donor T cells, in the case of allogeneic stem cell transplants, leading to their activation and inflammatory cytokine production [109].

Diffuse alveolar hemorrhage can also occur in autologous transplant recipients [107,109]. The T cells might be activated by compounds such as lipopolysaccharides, which end up in the bloodstream following gastrointestinal epithelium damage, as is the case in mucositis (caused by melphalan, a drug used in the treatment of multiple myeloma) or GVHD [107,109]. GVHD leads to alveolitis, manifesting as alveolar hemorrhage and increased counts of alveolar leukocytes, regardless of post-transplantation leukopenia [110]. Consequently, the endothelial swelling and medial hyperplasia leads to narrower vessel lumina, increasing the extravasation of erythrocytes into the lung parenchyma [110]. Outcomes for diffuse alveolar hemorrhage appear remarkably poor, with reported mortality rates between 64 and 100% [109,110]. Supportive therapy includes platelet transfusions, clotting factor (recombinant factor VIIa) or antifibrinolytic drug intake, and ventilatory support, while corticosteroids are largely unhelpful [109,110]. The recombinant factor VIIa, particularly when administered locally, through bronchoscopy, overcomes any preexisting tissue factor pathway inhibitors and leads to bleeding control [110].

Engraftment syndrome (ES) occurs at a reported rate of 7–90% in patients receiving HSCT, most often after autologous HSCTs [111]. ES occurs within 4 days of engraftment, which is defined as the first of 3 consecutive days when neutrophil levels maintain themselves at over 500/mm^3^, and it is caused by the engrafted neutrophils’ production of inflammatory cytokines and degranulation, leading to systemic endothelial damage [112,113]. A particular manifestation of ES is periengraftment respiratory distress syndrome, whose reported incidence rates vary from 2.5 to 25% [109]. The risk factors for PERDS are female gender, a quick immune function recovery, autologous HSCT, less intensive pre-conditioning therapy or GM-CSF instead of G-CSF, and the need of preengraftment platelet transfusions [109,114]. PERDS has a very similar clinical presentation to that of acute GVHD and, while self-limited, can be severe enough to warrant corticosteroid therapy (3 days of 1–2 mg/kg iv methylprednisolone q12h), along with supportive therapy [37,109].

### 3.6. TRALI in Patients with Acute Hematologic Malignancy

Since bone marrow suppression in hematological malignancies is either a consequence of the disease itself, or a desired effect of medication (as is the case in myeloablative therapy), cytopenia in at least one blood cell line occurs in almost all patients [115]. Once the levels of a particular component reach a critically low value, blood product transfusions should be performed [116,117]. The transfusion threshold for hemoglobin and platelets is 7 g hemoglobin/100 mL whole blood and 10 × 10^3^ platelets/mm^3^ whole blood, respectively (numbers apply in the absence of active bleeding) [116,117]. While their usefulness cannot be understated, transfusions are plagued by side effects, with transfusion-related acute lung injury (TRALI) being one of the most important issues when evaluating the prognosis of these patients.

TRALI is defined as pulmonary edema occurring within 6 h post-transfusion, in the absence of other ARDS-precipitating factors or evidence of circulatory overload [118]. It occurs in roughly 1 in every 1000 blood product recipients, with the incidence being 50 to 80 times higher in intensive care settings [118]. However, patients with hematologic malignancy develop this complication less frequently than other patient categories, most likely due to the associated neutropenia [118]. Mortality stands at approximately 10%, while mechanical ventilation requirement occurs in 70 to 90% of cases [119]. Risk factors associated with TRALI are the total number of administered blood products, previously pregnant donors, chronic alcohol and tobacco use, pretransfusion shock, and positive fluid balance [118,119]. The occurrence of TRALI can most often be attributed to the presence of leukocyte antibodies in the plasma contained in blood products [118]. The antibodies bind their corresponding recipient antigens and trigger an immune reaction culminating in resident neutrophil activation, capillary leak, and lung injury [118].

TRALI in neutropenic patients might occur due to antibodies binding directly to endothelial cells, which are then damaged by reactive oxygen species produced through the activation of the complement cascade or monocytes [118]. It has been proven that even platelets are capable of secreting proinflammatory mediators, while also migrating into alveolae, augmenting leukocytic infiltration [118]. Finally, erythrocyte transfusion bags might carry significant amounts of proinflammatory factors (reactive oxygen species, cytokines, etc.), which can trigger acute lung injury in the recipient [118,119].

TRALI is self-limited, with patients recovering within 3 to 4 days. In some cases, corticosteroids or diuretic therapy might prove useful [2]. In order to reduce the occurrence rate, mitigation strategies have been implemented: collecting plasma-rich products only from male donors or nulliparous females, antibody screening in female thrombocyte apheresis donors (since this product contains a significant amount of plasma), or pooling together plasma and platelets from multiple donors, to dilute or neutralize any residual antibodies [119].

## 4. Conclusions

ARDS in acute hematologic malignancy represents a real therapeutic challenge, mainly due to the etiological heterogeneity. Moreover, the short-term prognosis relies on prompt diagnosis and treatment. Further precision-based strategies aiming to overcome this heterogeneity should be developed, in the hope of aiding clinicians establishing a diagnosis more accurately and rapidly. Future studies should focus on identifying the relevance of such approaches in ARDS secondary to acute hematologic malignancy.

## Figures and Tables

**Figure 1 medicina-58-01215-f001:**
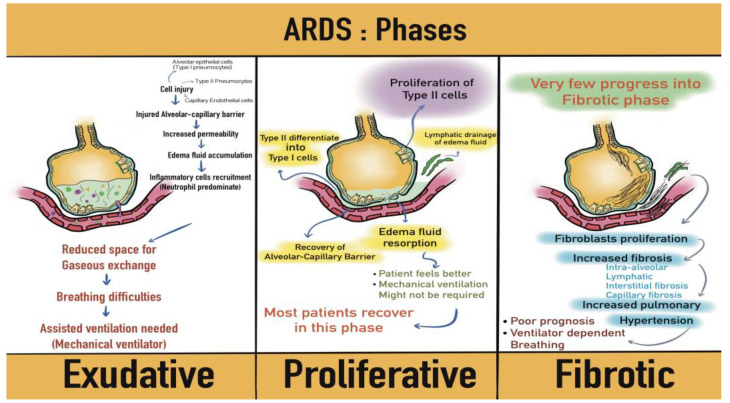
The progression of histopathological findings in acute respiratory distress syndrome (ARDS).

**Figure 2 medicina-58-01215-f002:**
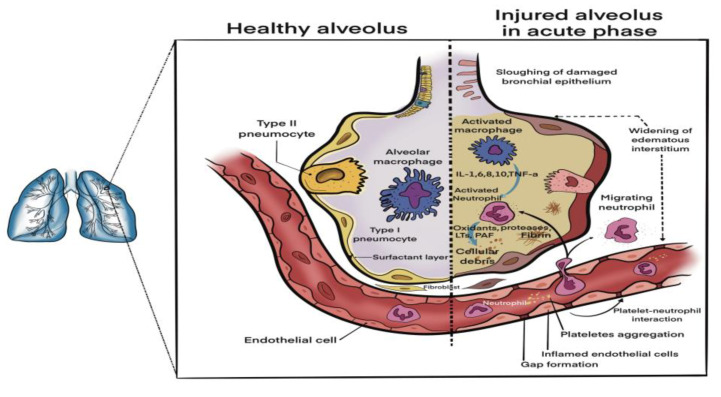
Biochemical changes in acute respiratory distress syndrome (ARDS)-damaged lung tissue.

## Data Availability

Not applicable.

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
