# Peer review of "What Is Different in Acute Hematologic Malignancy-Associated ARDS? An Overview of the Literature"

_medicina, 2022, doi:10.3390/medicina58091215_

Round 1
Reviewer 1 Report
The review written by Cotorega-Simon et al., addressing acute respiratory distress syndrome (ARDS) amongst patients with acute hematologic malignancies is a timely summary of current knowledge on the pathogenic mechanisms and therapeutic options for ARDS. Overall, the literature is well-reviewed, well written with appropriate conclusions.
Author Response
Response to Reviewer 1
(Manuscript id: 1860476)
We are very thankful to the editor for the opportunity to submit a revised form of our article.
The reviewer comments: The review written by Cotorega-Simon et al., addressing acute respiratory distress syndrome (ARDS) amongst patients with acute hematologic malignancies is a timely summary of current knowledge on the pathogenic mechanisms and therapeutic options for ARDS. Overall, the literature is well-reviewed, well written with appropriate conclusions
Response: We thank the reviewer for the appreciation.

Reviewer 2 Report
Minor editing may improve the sentence structure. for example: "The most relevant and potentially useful for clinical practice phenotypes are the so-called “hyperinflammatory” and “hypoinflammatory”, the former exhibiting higher levels of circulating proinflammatory interleukins (IL-6, IL-8, IL-10), PAI-1, soluble receptors 139 for TNF-α and advanced glycation end-products, and lower levels of C protein in the blood and surfactant proteins in the BALF (Fig. 2)." would sound more appropriate: " The phenotypes most relevant and potentially useful for clinical practice..." This awkward long sentence could also be reformulated in several shorter sentences.
Author Response
Response to Reviewer 2 Comments
(Manuscript id: 1860476)
We are very thankful to the editor for the opportunity to submit a revised form of our article. We hope our revision has improved the paper to an acceptable level for publication in Medicina.
Response to Reviewer 2 Comments
We thank this referee for the constructive comments and very useful suggestions.
Comments to the Author:
Minor editing may improve the sentence structure. for example: "The most relevant and potentially useful for clinical practice phenotypes are the so-called “hyperinflammatory” and “hypoinflammatory”, the former exhibiting higher levels of circulating proinflammatory interleukins (IL-6, IL-8, IL-10), PAI-1, soluble receptors 139 for TNF-α and advanced glycation end-products, and lower levels of C protein in the blood and surfactant proteins in the BALF (Fig. 2)." would sound more appropriate: " The phenotypes most relevant and potentially useful for clinical practice..." This awkward long sentence could also be reformulated in several shorter sentences.
Response: We thank the reviewer for the useful comment. We rechecked the manuscript and changed the sentence structures and spelling with track change function.
